# A solvent-assisted ligand exchange approach enables metal-organic frameworks with diverse and complex architectures

Dongbo Yu [1,2,7], Qi Shao[1,7], Qingjing Song[1], Jiewu Cui [1✉], Yongli Zhang[1], Bin Wu[3], Liang Ge[2], Yan Wang[1], Yong Zhang[1], Yongqiang Qin[1], Robert Vajtai [4,5], Pulickel M. Ajayan [4✉], Huanting Wang [6], Tongwen Xu [2✉] & Yucheng Wu[1✉]

Unlike inorganic crystals, metal-organic frameworks do not have a well-developed nanostructure library, and establishing their appropriately diverse and complex architectures remains a major challenge. Here, we demonstrate a general route to control metal-organic framework structure by a solvent-assisted ligand exchange approach. Thirteen different types of metal-organic framework structures have been prepared successfully. To demonstrate a proof of concept application, we used the obtained metal-organic framework materials as precursors for synthesizing nanoporous carbons and investigated their electrochemical $Na^+$ storage properties. Due to the unique architecture, the one-dimensional nanoporous carbon derived from double-shelled ZnCo bimetallic zeolitic imidazolate framework nanotubes exhibits high specific capacity as well as superior rate capability and cycling stability. Our study offers an avenue for the controllable preparation of well-designed meta-organic framework structures and their derivatives, which would further broaden the application opportunities of metal-organic framework materials.

[1] School of Materials Science and Engineering, Hefei University of Technology, Hefei 230009, P. R. China. [2] CAS Key Laboratory of Soft Matter Chemistry, Collaborative Innovation Center of Chemistry for Energy Materials, School of Chemistry and Material Science University of Science and Technology of China, Hefei 230026, P. R. China. [3] Key Laboratory of Environment-Friendly Polymeric Materials of Anhui Province, School of Chemistry & Chemical Engineering Anhui University, Hefei 230601, P. R. China. [4] Department of Materials Science and NanoEngineering, Rice University, Houston, TX 77005, USA. [5] University of Szeged, Interdisciplinary Excellence Centre, Department of Applied and Environmental Chemistry, University of Szeged, Rerrich Béla tér 1., Szeged, Hungary. [6] Department of Chemical Engineering, Monash University, Clayton, VIC 3800, Australia. [7] These authors contributed equally: Dongbo Yu, Qi Shao. ✉email: jwcui@hfut.edu.cn; ajayan@rice.edu; twxu@ustc.edu.cn; ycwu@hfut.edu.cn

Metal–organic frameworks (MOFs) are a class of porous inorganic–organic hybrid crystals[1], which have been employed in a wide range of applications, owing to their fascinating properties such as high surface area, structural and functional flexibility[2–4]. The study of MOFs has emerged as one of the most advanced research frontiers. To realize more targeted functionalities, substantial efforts mainly focus on the development of new types of MOFs[5,6], molecular level manipulations (e.g., chemical modification and aperture size adjustment)[1,7], and megascopic manufacturing (e.g., MOF membranes)[8]. By contrast, only a small number of attempts have been devoted to tailoring MOF architectures on the micro-/nano-scale, even if architectural design is known to have a vital effect on performance[9–11]. Some latest significant progresses actually benefit from very common architectures such as MOF@MOF core-shell structures and MOF hollow structures[9–18]. Nevertheless, unlike inorganic crystals which have abundant nanostructure libraries and well-developed preparation approaches[19,20], MOFs commonly lack nano-morphological diversity and nanostructural complexity, a large proportion of MOFs are solid particle shapes, which extremely limits the application of MOF materials.

Currently, most of the reported core-shell MOFs are prepared by an epitaxial growth method[21], and the shell MOFs often need to have the identical topology structure of the core MOFs[11,15,16]. In the light of crystal growth theory, nucleation favors a coherent phase boundary between the substrate and target crystal. To hybridize MOFs with different topology, a surfactant-mediated strategy is proposed[17,18], while there are still many core MOFs totally uncovered or partially covered by the shell MOFs. Another double hydrophilic block copolymer-modulated method is also demonstrated to design a core-shell MOF hybrid with relatively low crystallinity of MOF shell[22]. In addition, since the convex surface of one-dimensional (1D) MOFs is not as favorable for crystallization as the flat plane of polyhedral MOFs, 1D MOF@MOF core-shell nanostructures have rarely been reported. Different from core-shell structure, hollow structure possesses a specific enclosed space, representing another very important configuration[10]. To date, simple single-shelled hollow particles dominate the majority of hollow MOFs, as some more complex hollow MOFs start to appear in recent reports[13,14]. However, the successful fabrication of these complex structures involves complicated growth-and-etching procedures and has various extreme prerequisites; each of these methods is unlikely to be extended to other MOFs and is solely for a specific MOF. Therefore, given that the study of MOF nanostructure regulation is in its infancy, the researchers have sufficient motivation to develop new, facile and versatile synthesis strategies in order to enrich the MOF nanostructure library.

In this manuscript, we successfully fabricate thirteen different MOF nanoarchitectures via a developed solvent-assisted ligand exchange (SALE) approach[23–26], although SALE is conventionally used for MOF-related functional modifications. Considering that, in theory, the thermodynamic driving force during the exchange process is the formation of more stable coordination bonds[27], we thus choose stable zeolitic imidazolate frameworks (ZIFs) as target daughter MOFs, and 1D/2D/3D MOFs with weaker bonds serve as the mother MOFs (Fig. 1). Because a typical SALE transformation generally involves two interweaving steps of dissolution and recrystallization[28], through precisely adjusting the balance between the cleavage of old bonds and the establishment of new bonds, we can obtain different kinds of 1D/2D/3D MOF nanostructures, as shown in Supplementary Fig. 1. These architectures cover most of the nanostructures in inorganic crystals. Very encouragingly, 1D double-shelled nanotube, nanowire-nanotube and peapod-like MOF structures have been synthesized. It is reported that some MOFs can be directly used for electrochemical applications[29], but

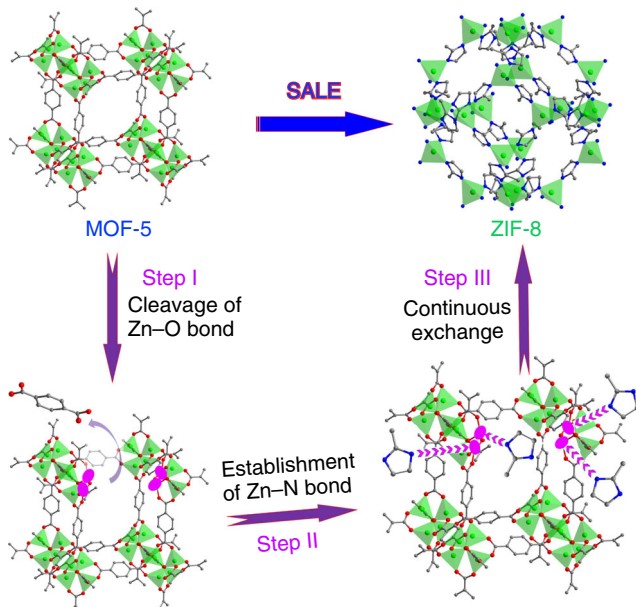

**Fig. 1 Schematic representation.** A typical solvent-assisted ligand exchange (SALE) process with the transformation from MOF-5 to ZIF-8 as an example (gray ball: C, red ball: O, blue ball: N, green ball: Zn), the pink elliptic regions denoted unsaturated $Zn^{2+}$ nodes after the cleavage of Zn-O coordination.

the relatively high internal resistance and poor stability truly retards the further promotion of performance. MOF-derived nanoporous carbon demonstrates a feasible option, which can integrate the porous nature and nano-architectures of pristine MOF precursors with excellent conductivity and stability of carbon by a simple carbonization treatment[30]. In view of this, as a potential application of these MOF nanostructures, the as-prepared MOFs are subsequently carbonized to fabricate nanoporous carbon for $Na^+$ ion storage. Electrochemical tests indicate that due to the specific architectures inherited from MOF precursors, these nanoporous carbon materials exhibit much better electrochemical energy storage properties than that of common ZIF-8 solid particle-derived carbon, and double-shelled ZnCo bimetallic ZIF nanotube-derived nanoporous carbon delivers very competitive performance compared to previously reported carbon materials. Our present work demonstrates a pioneering study to schematically refine the management of MOF nanostructures. The strategy we propose here can be applied more generally to other MOFs, which offers the strong possibility of realizing a massive selection of MOF nanostructures, hence extending their utilization scope remarkably.

## Results

**Synthesis and formation mechanism of 3D MOF architectures.** MOF-5, assembled by $Zn_4O^{6+}$ clusters and $BDC^{2-}$ (BDC = 1,4-benzenedicarboxylate) ligands, is unstable even under atmospheric conditions because its weak metal-oxygen coordination can be disrupted by water via ligand displacement[31]. The as-synthesized MOF-5 had a hexahedral shape and smooth surface (Supplementary Fig. 2a). When MOF-5 was placed in 1.2 M 2-methylimidazole (Hmim) ethanol solution at room temperature overnight, the surface of the resulting material became rougher, and hollow features were observed in some broken nanocubes (Supplementary Fig. 2b). The enlarged scanning electron microscopy (SEM) image revealed that such hollow nanocubes possessed a micro-nano superstructure that was composed of many stacked nanoparticles (Supplementary Fig. 2c), implying that the transformation was not controlled by the epitaxial growth

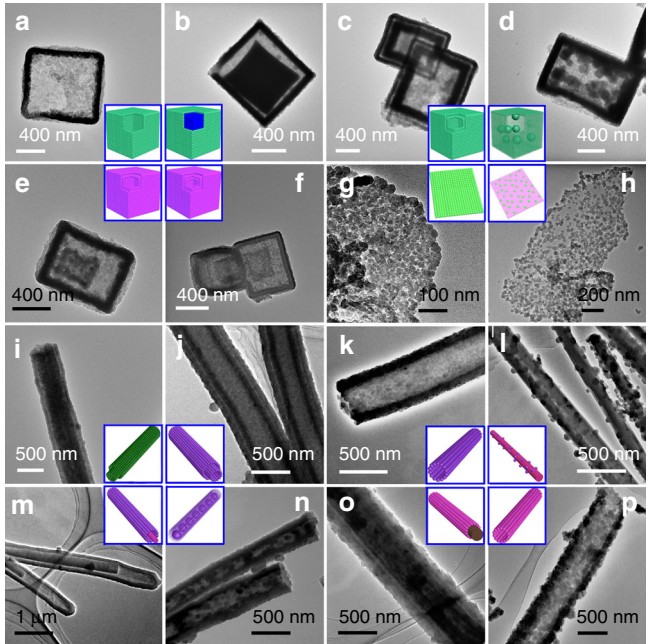

**Fig. 2 Zeolitic imidazolate framework-based nanoarchitectures.** TEM images of SH-ZIF-8 nanocube (**a**), yolk-shell MOF-5@ZIF-8 nanocube (**b**), DH-ZIF-8 nanocube (**c**), ball-in-box ZIF-8 nanocube (**d**), DH-ZnCo-ZIF nanocube (**e**), TH-ZnCo-ZIF nanocube (**f**), ZIF-8 nanosheet (**g**), sesame pancake-like Zn-HMT@ZIF-8 nanosheet (**h**), DT-ZIF-8 nanotube (**i**), DT-ZnCo-ZIF nanotube (**j**), ST-ZnCo-ZIF nanotube (**k**), bead-on-string hybrid structure (**l**), nanowire-nanotube hybrid structure (**m**), peapod-like structure (**n**), ZnCo-MOF-74@ZnCo-ZIF core-shell structure (**o**) and ZnCo-MOF-74-derived ZnCo-ZIF nanotube (**p**).

mechanism. Transmission electron microscopy (TEM) images further confirmed the single-shelled structure (Fig. 2a, Supplementary Fig. 2d), and the corresponding energy dispersive X-ray spectroscopy (EDX) mapping images indicated a uniform element distribution (Supplementary Fig. 2e). The X-ray diffraction (XRD) pattern of the hollow nanocubes matched well with the characteristic diffraction peaks of ZIF-8, and no impurity peak was detected (Supplementary Fig. 2f). Moreover, Fourier transform infrared (FT-IR) spectra clearly clarified the coordination between deprotonated Hmim (mim$^-$) and Zn$^{2+}$ (Supplementary Fig. 3). Consequently, single-shelled hollow ZIF-8 nanocubes (SH-ZIF-8) were prepared. By simply shortening the time to 2 min, it was easy for us to obtain the MOF-5@ZIF-8 yolk-shell structure (Fig. 2b, Supplementary Fig. 4). If the temperature was increased to 45 °C, MOF-5 evolved to double-shelled hollow ZIF-8 (DH-ZIF-8) (Fig. 2c, Supplementary Fig. 5).

The transformation of these MOF-5-derived structures actually results from a diffusion-controlled exchange process[28]. In terms of our previous studies, Hmim solution could induce a weakly acidic environment (Hmim $\rightleftharpoons$ (1-$x$)H$^+$ + (1-$x$)mim$^-$ + $x$Hmim), which breaks ionic bonds such as Co-OH and Zn-OH[32,33]. Accordingly, one can reason that the weaker Zn-O tetrahedral coordination of MOF-5 would be deteriorated once exposed to such an environment, and Zn$^{2+}$ nodes are in an unsaturated state (Step I in Fig. 1). Meanwhile, deprotonated mim$^-$ is prone to coordinating with unsaturated Zn$^{2+}$ and establishing new more stable Zn-N tetrahedral coordination[31], which completes one exchange cycle between BDC$^{2-}$ and mim$^-$ (Step II in Fig. 1). With continuous ligand exchange, each unit cell finally transforms from MOF-5 to ZIF-8 (Step III in Fig. 1), and ZIF-8 starts to nucleate on the surface of MOF-5 (Supplementary Fig. 6). The ZIF-8 nanocrystals gradually generate with the time, until a thin ZIF-8 layer compactly

covers MOF-5 to produce a core-shell hybrid structure, the ion/molecule transfer is restricted, and the SALE process is diffusion-controlled (Supplementary Fig. 7). In this stage, only a limited number of mim$^-$ can diffuse inward to create a gradually descending molar ratio of mim$^-$/Zn$^{2+}$ from outside to inside, the ZIF particle size would increase in order[34], and thus the thickness from the outer shell to the inner shells increases certainly (Supplementary Fig. 7). On the one hand, even at a very low concentration of mim$^-$, the ligand can readily disrupt the weak coordination bond of MOF-5; on the other hand, the formation of stable ZIF-8 nuclei requires a certain molar ratio of mim$^-$/Zn$^{2+}$,[34]. As a result, the recrystallization rate of ZIF-8 is slower than the dissolution rate of MOF-5. The non-equivalent diffusion gives rise to considerable voids near the interfaces of MOF-5/ZIF-8 in a short time, rightly yielding the yolk-shell structure (Fig. 2b, Supplementary Fig. 6, Supplementary Fig. 8).

It is well known that higher temperature leads to wider vibration amplitude of frames and larger spatial change[35]. The increasing temperature facilitates more effective diffusion of mim$^-$ ligands for crystallizing secondary ZIF-8 shell evolved from yolk MOF-5, finally DH-ZIF-8 and even the triple-shelled hollow structure (Supplementary Fig. 7e-h) are prepared at 45 °C. At room temperature, the amount of transferred mim$^-$ ligands is insufficient to render the crystallization of ZIF-8 on yolk MOF-5 surface immediately; simultaneously, by reason of the trend of reducing nucleation energy and the relatively higher mim$^-$/Zn$^{2+}$ ratio nearby the shell wall than in the center, the growth of ZIF-8 thus prefers to continue on the inner wall of ZIF-8 shell, leading to single-shelled hollow structure with thicker wall (Fig. 2a, Supplementary Fig. 7a, b). When lower concentrated Hmim solution is used, the coordination bond of MOF-5 can be quickly cleaved, while the ZIF-8 nucleation would only happen by the time the mim$^-$/Zn$^{2+}$ ratio reaches a certain value; in this case, it is preferable to grow large-sized ZIF-8 inside the shell, and we consequently obtain an interesting ball-in-box ZIF-8 architecture (Fig. 2d, Supplementary Fig. 6o, Supplementary Fig. 9). If the concentration is further diluted to 0.15 M, without adequate Hmim ligands for the crystallization of ZIF-8, the product is actually an inhomogeneous mixture containing metastable metal-organic coordination compounds[36], and the transformation from MOF-5 to ZIF-8 is incomplete (Supplementary Fig. 10a-d). In very concentrated solutions, big cracks can be observed on ZIF-8 nanocubes (Supplementary Fig. 10e-h), owing to the stress concentration generated from such violent phase transition. As a consequence, a proper concentration range is needed not only to drive a rational kinetics for crystallizing ZIF-8, but also to offer steady ligand exchange for inheriting the shape of MOF-5 nanocubes; a higher concentration helps with faster kinetics for phase separation to yield yolk-shell structure, and facilitates greater mim$^-$ concentration gradient along the vertical direction of shell wall afterwards, which is in favor of larger mim$^-$/Zn$^{2+}$ ratio inside the hollow interior; the mim$^-$/Zn$^{2+}$ ratio finally determines the evolution behavior of MOF-5 core as discussed above (Supplementary Fig. 6), and we obtain SH-ZIF-8 and ball-in-box structure. It is noted that very high mim$^-$ concentration might also enable sufficient diffused mim$^-$ ligands in the enclosed shell, forming secondary ZIF-8 shell derived from the internal yolk (mentioned below in ZIF-71-derived double-shelled hollow ZIF-8), which is broadly equivalent to the effect of increasing temperature. Furthermore, the dissolution of MOF-5 should keep in step with the recrystallization of ZIF-8[37], otherwise, if the dissolution process is too fast to provide available substrate for nucleation, taking for example, water as the solvent, it generates monodispersed ZIF-8 particles (Supplementary Fig. 11).

Heterobimetallic MOFs have attracted increasing attention recently[12,38], we here also prepared Zn/Co-containing ZIF

(ZnCo-ZIF) using Co-doped MOF-5 (ZnCo-MOF-5) as the mother MOF. Both $Zn^{2+}$ and $Co^{2+}$ ions are known to exist in either tetrahedral or octahedral coordination modes; however, in the presence of weak-field ligands such as carboxylates and solvent molecules, $Zn^{2+}$ favors tetrahedral coordination, while $Co^{2+}$ prefers octahedral coordination[39]. Therefore, it is easier to rebuild Zn-N tetrahedral coordination (for ZIF-8) than Co-N tetrahedral coordination (for ZIF-67) in a Hmim solution, and theoretically, the recrystallization rate of ZnCo-MOF-5-derived ZnCo-ZIF is lower than that of MOF-5-derived ZIF-8. In our experiment, if the reaction proceeded at 45 °C, the cubic shape of ZnCo-MOF-5 was lost, and the final products were microspheres with sizes from tens of nanometers to several micrometers (Supplementary Fig. 12). We thus carried out the reaction in a 0 °C icebox to slow down the reaction kinetics. Single-shelled (Supplementary Fig. 13), double-shelled (Fig. 2e, Supplementary Fig. 14) and triple-shelled (Fig. 2f, Supplementary Fig. 14) hollow ZnCo-ZIF structures could be achieved when carefully manipulating the reaction kinetics similar to MOF-5-derived ZIF-8 hollow structures. Apart from time, temperature, concentration, solvent type and heteroatom, the type of mother MOF inherently affects bond energy and thus the cleavage of coordination bonds. ZIF-8, ZIF-7 and ZIF-71 have similar Zn-N tetrahedral coordination and SOD topology, but their linker $pK_a$ values follow the trend $pK_a$ (ZIF-8) > $pK_a$ (ZIF-7) > $pK_a$ (ZIF-71). The higher $pK_a$ is of the linker substituent, the better the stability of the coordination bond[31,40,41]. Our experimental results verify the theoretical expectation that the transformation from ZIF-7 to ZIF-8 requires more severe conditions to overcome the higher reaction barrier in comparison with the cases of ZIF-71 and MOF-5. In addition, ZIF-7-derived single-shelled hollow ZIF-8 (Supplementary Fig. 15), ZIF-71-derived single-shelled (Supplementary Fig. 16) and double-shelled (Supplementary Fig. 17) hollow ZIF-8 are fabricated naturally.

**Synthesis of 2D MOF architectures**. Based on encouraging results from the nanostructure regulation of 3D MOFs, we generalize this strategy to 1D and 2D MOF nanoarchitectures. Zn-hexamine coordination framework (Zn-HMT) nanosheets are selected as the 2D mother MOF[42]. Instead of water, ethanol, N,N-dimethylformamide and N-methyl pyrrolidone, methanol was found to be the best solvent for the successful synthesis of ZIF-8 nanosheets, indicating that solvent type plays a crucial role in the SALE process (Supplementary Fig. 18). Placing Zn-HMT in 0.75 M and 0.25 M Hmim methanol solutions for 15 min, 2D ZIF-8 nanosheets (Fig. 2g, Supplementary Fig. 19) and sesame pancake-like Zn-HMT@ZIF-8 nanosheets (Fig. 2h, Supplementary Fig. 20) were obtained, respectively. We did not obtain a 2D core-shell nanostructure because of the excessive thinness of the Zn-HMT nanosheets. The thickness of ZIF-8 nanosheets increased compared to Zn-HMT nanosheets, and each ZIF-8 nanosheet exhibited a 2D stacking architecture of ZIF-8 nanaoparticles (Supplementary Fig. 19e). The growth of ZIF-8 nanocrystals first started at the edge of Zn-HMT nanosheets where more unsaturated coordination bonds were exposed (Supplementary Fig. 20c-e). As mentioned that the coordination stability of mother MOF influences the ligand exchange. When a very stable 2D porphyrin paddlewheel framework-3 MOF nanosheet was used as the mother MOF[43], it failed to fabricate ZnCo-ZIF nanosheets (Supplementary Fig. 21).

**Synthesis of 1D MOF architectures**. As the convex surface and small radii of curvature is detrimental for interfacial bonding during subsequent heterogeneous nucleation and growth, reports on diverse and complex 1D MOF nanoarchitectures have seldom been seen so far. We find that the direct transformation from mother MOFs to daughter MOFs can address this issue well. 1D M-BTC (M = Co, Zn; BTC = 1,3,5-benzenetricarboxylate) nanowires were employed to establish 1D ZIF-based nanoarchitectures[44]. In the presence of 1.2 M Hmim ethanol/water mixed solution (volume ratio of $V_{ethanol}$:$V_{water}$ was set at 9:1) at 45 °C for 2 h, Zn-BTC nanowires converted to double-shelled ZIF-8 nanotubes (DT-ZIF-8, Fig. 2i, Supplementary Fig. 22). The formation of double-shelled nanotube structure was virtually similar to those of 3D ZIF double-shelled hollow structures, the only difference between them was that 1D configuration caused confining diffusion along the axis direction. With the continuous and fast accumulation of vacancies at the ZIF-8/Zn-BTC phase interface by a non-equivalent diffusion effect, phase separation would take place; the initially generated ZIF-8 layer on Zn-BTC nanowire would grow into the outer nanotube, and the inner nanotube finally evolved from the unreacted Zn-BTC nanowire. The very thick inner nanotube wall of each DT-ZIF-8 was identical to the inner shell wall of MOF-5-derived DH-ZIF-8, and the hollow feature of the inner tube was not as obvious. Likewise the involvement of $Co^{2+}$ created more moderate ligand exchange conditions, which deferred the vacancy clustering and slowed down the ZnCo-ZIF crystallization, resulting in the delaying phase separation. It was interesting to see the thickness of the inner and the outer tube walls for double-shelled ZnCo-ZIF nanotubes (DT-ZnCo-ZIF, Fig. 2j, Supplementary Fig. 23) was almost the same when a bit of $Co^{2+}$ was introduced (Fig. 2j, Supplementary Fig. 23, Supplementary Fig. 24a, b). With the increasing $Co^{2+}$ content, the phase separation might not happen; or the ligand transfer was hindered by the thick outer tube wall, the low molar ratio of $mim^-$/$Zn^{2+}$ was not able to crystallize the secondary ZIF-8 shell, thus single-shelled ZnCo-ZIF nanotubes (ST-ZnCo-ZIF) were formed (Fig. 2k, Supplementary Fig. 25). We also observed some intriguing exceptions (Supplementary Fig. 24c-f); a triple-shelled trend emerged at the tips of some nanotubes (Supplementary Fig. 24d), and in particular, a few triple-shelled nanotubes were observed (Supplementary Fig. 24e-f). By adjusting the reaction time, bead-on-string (Fig. 2l, Supplementary Fig. 26), core-shell (Supplementary Fig. 27) and nanowire-nanotube (Fig. 2m, Supplementary Fig. 28) hybrid structures were prepared. The floccules in the nanowire-nanotube hybrids were considered as the intermediate species of ZnCo-ZIF (Supplementary Fig. 28d). If nanowire-nanotube hybrids reacted with Hmim again, we obtained a peapod-like ZnCo-ZIF structure (Fig. 2n, Supplementary Fig. 29) rather than DT-ZnCo-ZIF, suggesting a very special environment formed in the confined space of the nanotubes. These prepared 1D double-shelled nanotube, nanowire-nanotube and peapod-like MOF structures are also scarce for inorganic crystals and cannot be fabricated in a traditional way (Supplementary Fig. 30). Similarly, the use of more stable ZnCo-MOF-74 as the mother MOF yields core-shell hybrid (Fig. 2o, Supplementary Fig. 31) and single-shelled nanotube structures (Fig. 2p, Supplementary Fig. 32) which have an obviously grainy surface[45], and the particle size of ZnCo-ZIF is larger due to the high Co content of the mother MOF, whereas excess Co causes the loss of morphology control (Supplementary Fig. 33).

**Fabrication and characterization of nanoporous carbon**. Now, we have fabricated 21 different MOF materials and designed 13 different types of MOF architectures which cover most of the configurations that emerge in inorganic crystals; the proposed approach shows big advantages of versatility, simplicity, large-yield product and reproducibility. The synthesis routes of these MOF materials and structures all stem from the same SALE

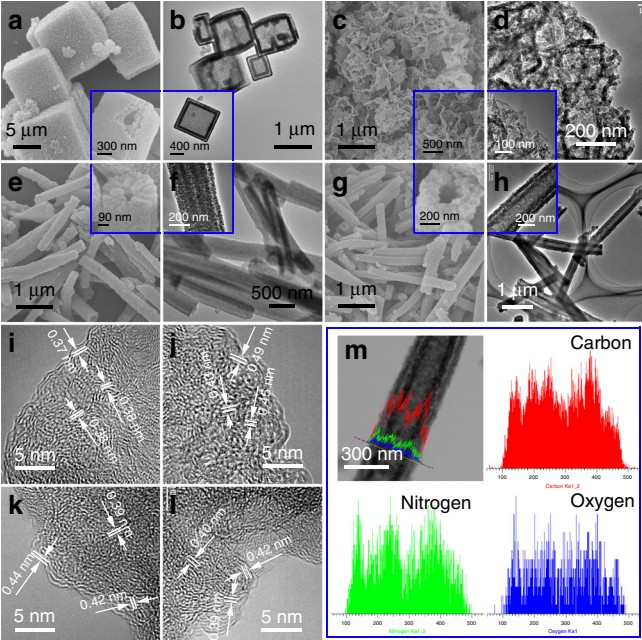

**Fig. 3 Zeolitic imidazolate framework-derived nanoporous carbon.** SEM (**a**) and TEM (**b**) images of 3D DH-Zn-ZIF-C; SEM (**c**) and TEM (**d**) images of 2D NS-Zn-ZIF-C; SEM (**e**) and TEM (**f**) images of 1D DT-Zn-ZIF-C; SEM (**g**) and TEM (**h**) images of 1D DT-ZnCo-ZIF-C; high-resolution TEM images of 3D DH-Zn-ZIF-C (**i**), 2D NS-Zn-ZIF-C (**j**), 1D DT-Zn-ZIF-C (**k**) and 1D DT-ZnCo-ZIF-C (**l**); EDX linear scanning images of 1D DT-ZnCo-ZIF-C (**m**).

strategy, which is a rare and commendable achievement. In this paper, we do not demonstrate the overall application potentials of these MOF architectures but take electrochemical Na$^+$ storage as an example. Due to the poor conductivity and instability, the as-prepared MOFs are not directly used as the electrode materials. Instead, a carbonization treatment is conducted to obtain MOF-derived nanoporous carbon. 3D DH-ZIF-8 nanocubes, 2D ZIF-8 nanosheets, 1D DT-ZIF-8 nanotubes and 1D DT-ZnCo-ZIF nanotubes were carbonized at 900 °C for 1 h to prepare the corresponding porous carbon materials, namely, DH-Zn-ZIF-C (Fig. 3a, b), NS-Zn-ZIF-C (Fig. 3c, d), DT-Zn-ZIF-C (Fig. 3e, f) and DT-ZnCo-ZIF-C (Fig. 3g, h), respectively. In addition, a solid ZIF-8 particle-derived porous carbon counterpart (s-Zn-ZIF-C) was prepared for comparison (Supplementary Fig. 34). The porous carbon materials basically maintained the initial morphology of their precursor MOFs, and there was no severe fracture or agglomeration; only distinct shrinkage could be observed. The C(002) diffraction peaks for DH-Zn-ZIF-C and NS-Zn-ZIF-C were located at the highest and lowest 2θ angles, respectively (Supplementary Fig. 35). The high-resolution TEM images (Fig. 3i–l) also verified the graphitization degree of four ZIF-derived carbon materials; DH-Zn-ZIF-C had the smallest interplanar spacing of C(002) (0.37–0.38 nm) and the highest graphitization degree in comparison with NS-Zn-ZIF-C (0.43–0.49 nm), DT-Zn-ZIF-C (0.39–0.44 nm) and DT-ZnCo-ZIF-C (0.39–0.42 nm). As the previous study demonstrated that a critical minimum spacing of 0.37 nm was often required for good Na$^+$ insertion properties[46], the enlarged interlayer distance would not only facilitate Na$^+$ ion diffusion[47], but also improve the activation polarization during the Na$^+$ intercalation/deintercalation processes. EDX mapping characterization verified the uniform N-doping in these ZIF-derived carbons (Fig. 3m, Supplementary Fig. 36). The N-doping content of s-Zn-ZIF-C,

DH-Zn-ZIF-C, NS-Zn-ZIF-C, DT-Zn-ZIF-C and DT-ZnCo-ZIF-C was finally determined to be 11.11%, 3.69%, 13.34%, 9.30% and 8.47%, respectively (Supplementary Fig. 37b-e, Supplementary Table 1) by X-ray photoelectron spectroscopy (XPS). Due to the different electron configuration of N in the carbon lattices, graphitic N was inert to Na$^+$; in contrast, both pyridinic N and pyrrolic N were very active in capturing Na$^+$ ions for electrochemical energy storage, and the induced electrostatic repulsion effect resulted in the expansion of the interlayer distance[47]. Hence, the largest C(002) interplanar spacing value, observed for NS-Zn-ZIF-C, could be attributed to this material containing the maximum amount of pyrrolic and pyridinic N-doping. The G-band shift in Raman spectra further confirmed the above XRD and XPS analyses (Supplementary Fig. 37f). The specific surface area and pore size distribution were evaluated by the nitrogen sorption technique (Supplementary Fig. 38), and 2D NS-Zn-ZIF-C showed the lowest Brunauer-Emmett-Teller (BET) surface area value of 38.4 m$^2$ g$^{-1}$, while 3D s-Zn-ZIF-C achieved the highest BET value of 1237.2 m$^2$ g$^{-1}$. The Barrett-Joyner-Halenda pore size distribution plots of DH-Zn-ZIF-C and NS-Zn-ZIF-C displayed a trend for pore sizes larger than 10 nm. Both DT-Zn-ZIF-C and DT-ZnCo-ZIF-C showed a relatively narrow pore size distribution (<10 nm), and most of their average pore sizes ranged from ~1–2 nm, although DT-ZnCo-ZIF-C had more ~2–4 nm mesopores. Micropores and mesopores might result from the inheritance of ZIF frameworks, the evaporation of Zn and the removal of Co/Zn species by acid leaching, while the generation of macropores could be originated from the severe shrinkage of ZIF nanoparticle-stacking during carbonization[48,49]. Note that the hierarchical porous architecture significantly affects Na$^+$ storage performance[50]. The micropores in porous carbon can offer more active sites to contribute to Na$^+$ adsorption and thus the specific capacity, while the mesoporous and macroporous structures chiefly serve as interconnected pathways for accessible ion transfer and buffer the volume change in the charge-discharge process for good cycling stability.

**Electrochemical Na$^+$ storage performance.** The electrochemical Na$^+$ storage properties of ZIF-derived porous carbon were estimated by assembling 2032 type coin cells. Figure 4a shows the cyclic voltammetry (CV) plots of DT-ZnCo-ZIF-C; a broad peak located at ~0.01–1.0 V can be identified in the 1$^{st}$ cycle, which is attributed to the formation of solid electrolyte interface (SEI) film[42]. The 3$^{rd}$ CV curve nearly completely overlaps the 2$^{nd}$ cycle, suggesting an excellent reversible Na$^+$ storage property of DT-ZnCo-ZIF-C. The galvanostatic charge-discharge profiles of the DT-ZnCo-ZIF-C electrode were recorded in the potential range of ~0–3 V at a current density of 0.1 A g$^{-1}$ (Fig. 4b). The charge and discharge capacities in the first cycle are 465 and 625 mAh g$^{-1}$, respectively, indicating an initial coulombic efficiency of 74.4%, and the irreversible capacity loss is caused by the formation of the SEI film and the decomposition of electrolyte[51]. DT-ZnCo-ZIF-C delivers the highest specific capacity among all the porous carbon electrodes, and 3D s-Zn-ZIF-C shows the most inferior Na$^+$ storage performance (Fig. 4c). At current densities of 0.05, 0.1, 0.2, 0.5, 1, 2, 5 and 10 A g$^{-1}$, DT-ZnCo-ZIF-C exhibits reversible specific capacities of 474, 455, 432, 396, 364, 305, 216 and 178 mAh g$^{-1}$, respectively. When the current density is returned to 0.1 A g$^{-1}$, an enhanced capacity of 490 mAh g$^{-1}$ is achieved; the possible reason for this phenomenon is that electrode activation occurs during the rate cycling so that full penetration of the electrodes is realized to afford more active sites for Na$^+$ storage[47]. We also observe that the high-rate performance of DT-Zn-ZIF-C is worse than that of NS-Zn-ZIF-C although DT-Zn-ZIF-C has a higher specific capacity at low

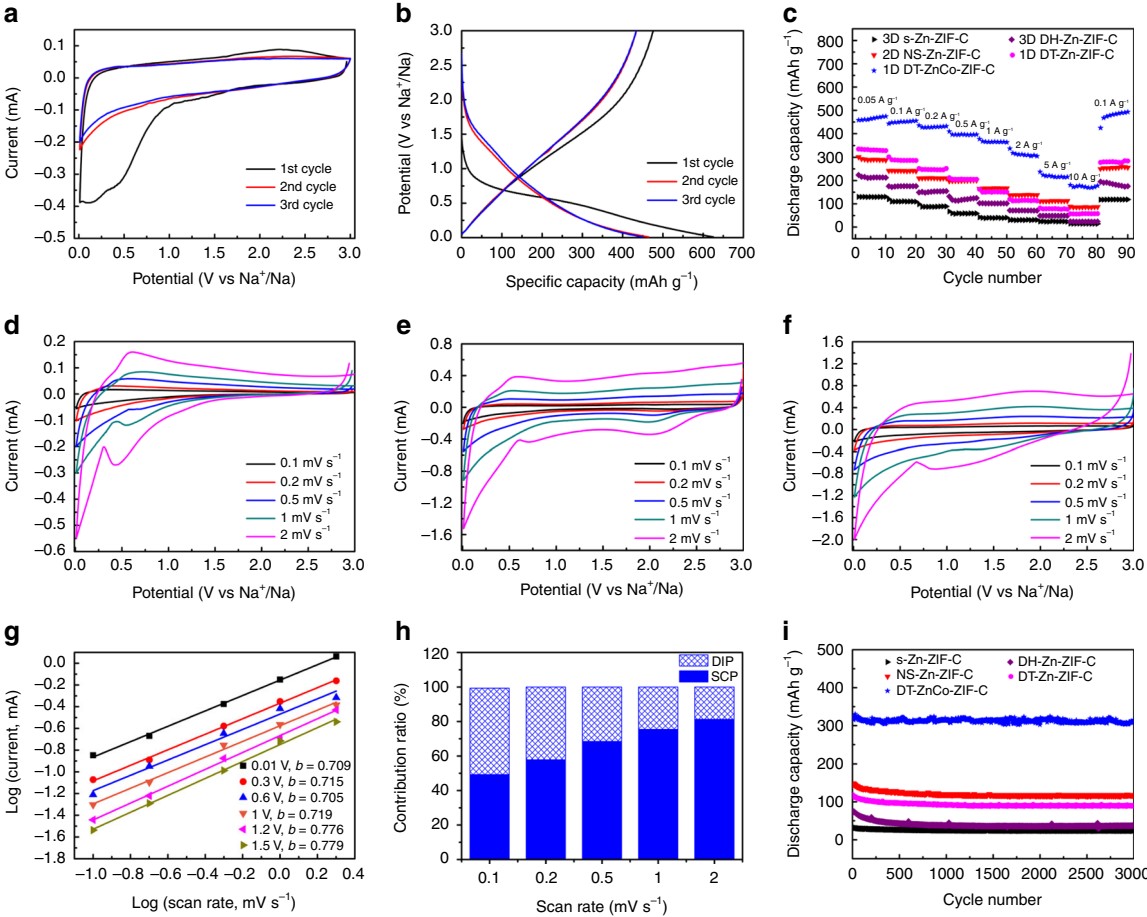

**Fig. 4 Electrochemical properties of the prepared nanoporous carbon.** CV curves at a scan rate of 0.1 mV s$^{-1}$ (**a**) and charge-discharge voltage profiles at a current density of 0.1 A g$^{-1}$ (**b**) of DT-ZnCo-ZIF-C; rate capability of 3D s-Zn-ZIF-C, 3D DH-Zn-ZIF-C, 2D NS-Zn-ZIF-C, 1D DT-Zn-ZIF-C and 1D DT-ZnCo-ZIF-C (**c**); CV plots of 3D DH-Zn-ZIF-C (**d**), 2D NS-Zn-ZIF-C (**e**) and 1D DT-ZnCo-ZIF-C (**f**) at different scan rates; the linear relationships between the logarithm current and logarithm scan rate (**g**) and the contribution ratio of DIP and SCP versus scan rate (**h**) calculated from CV curves of 1D DT-ZnCo-ZIF-C; long-term cycling performance at a current density of 2 A g$^{-1}$ (**i**).

current density; in addition, DH-Zn-ZIF-C manifests only a slightly larger capacity than s-Zn-ZIF-C at 10 A g$^{-1}$; furthermore, s-Zn-ZIF-C exhibits the most inferior Na$^+$ storage properties.

To obtain a better understanding of how the abovementioned different behaviors affect electrochemical Na$^+$ storage, CV curves of all ZIF-derived porous carbon are measured at various scan rates (Fig. 4d–f, Supplementary Fig. 39). According to previous studies, the relationship between current ($i$) and scan rate ($v$) can be described by the equation $i = av^b$, where a and b are adjustable parameters[52]. The Na$^+$ storage behavior is determined by the $b$ value. Generally, when $b = 0.5$, the electrochemical storage kinetics are subject to a diffusion-controlled intercalation process (DIP); when $b = 1$, the kinetics are controlled by a surface-induced capacitive process (SCP). The b values of DT-ZnCo-ZIF-C at ~0–1.5 V are within the range of ~0.7–0.78 (Fig. 4g), revealing the combined effect of the battery-type behavior and capacitive effect. Furthermore, the contribution ratio of DIP and SCP to the total capacity can be quantitatively calculated by the equation $i = k_1 v + k_2 v^{1/2}$, in which $i$ represents the current at a certain potential, and $k_1 v$ and $k_2 v^{1/2}$ indicate contributions from DIP and SCP, respectively[53]. The SCP contribution ratio of DT-ZnCo-ZIF-C increased from 49.3% to 81.3% with the increasing scan rate (Fig. 4h). In contrast, capacitive behavior (SCP) dominates the Na$^+$ storage process for NS-Zn-ZIF-C, while the major contribution to capacity for s-Zn-ZIF-C, DH-Zn-ZIF-C and DT-Zn-ZIF-C comes from DIP process (Supplementary

Fig. 40), which reasonably explains the inferior rate capacity. In addition to CV analysis, electrochemical impedance spectroscopy (EIS) is also analyzed. Each Nyquist plot (Supplementary Fig. 41) consists of a semicircle in the high-frequency region and a straight line in the low-frequency region, which respectively represents the charge transfer ability and ion transport efficiency. The larger diameter of the semicircle for 3D s-Zn-ZIF-C and DH-Zn-ZIF-C carbon particles corresponds to the higher ohmic resistance relative to 1D and 2D carbon nanostructures, and the worse conductivity endows the electrode with higher ohmic polarization. In addition, the steeper slope of the straight line for 2D NS-Zn-ZIF-C and 1D DT-ZnCo-ZIF-C indicates more effective Na$^+$ diffusion in comparison with 3D s-Zn-ZIF-C, 3D DH-Zn-ZIF-C and 1D DT-Zn-ZIF-C. As is known, during the charge-discharge process, the battery always suffers from battery polarization, which can be divided into ohmic polarization, activation polarization and concentration polarization. Although NS-Zn-ZIF-C had a very low surface area, its excellent conductivity improves the ohmic polarization, and it has the largest interlayer distance of the C(002) plane, which mitigates the negative influence of activation polarization and concentration polarization, resulting in an extraordinary rate capability. In contrast, s-Zn-ZIF-C has a high surface area and N-doping, but its solid particle structure causes the greatest ohmic resistance and blocks the Na$^+$ transport pathway, causing it to exhibit the lowest specific capacity and poor rate capability. All of the ZIF-derived

nanoporous carbon show better electrochemical properties than the common solid ZIF-8 particle-derived counterpart, suggesting the significance of rational design of MOF architectures. Stemming from the unique double-shelled hollow nanotube structure, DT-ZnCo-ZIF-C takes advantage of a relatively high surface area and N-doping for active sites, a favorable pore distribution for ion diffusion and good conductivity for charge transfer; as a result, the highest specific capacity and superior rate capability could be realized. More importantly, DT-ZnCo-ZIF-C retained 96.6% of the original value after 3000 repeated cycles at $2 \, A \, g^{-1}$, demonstrating outstanding cycling stability (Fig. 4i). In addition, the as-synthesized DT-ZnCo-ZIF-C also shows better or comparably competitive $Na^+$ storage performances compared to other reported MOF-derived carbon as well as other carbon materials (Supplementary Tables 2 and 3)[42,47,51,54–69].

## Discussion

In summary, we have developed a versatile strategy for the design of diverse and complex MOF architectures. By precisely controlling the reaction kinetics to adjust the balance between the cleavage rate of old bonds and the formation rate of new bonds, 21 different MOF materials and 13 different types of MOF configurations ranging from 3D to 2D and 1D structures have been successfully synthesized, covering almost all of the usual inorganic crystal configurations. As a proof-of-concept application, DT-ZnCo-ZIF-C as a $Na^+$ ion battery electrode demonstrates remarkable energy storage performance, which can be ascribed to the unique double-shelled nanotube architecture. More importantly, SALE is expected as a powerful strategy to offer unlimited possibilities for the design of other MOFs with various architectures.

## Methods

**Synthesis of 3D Zn-MOF-5/ZnCo-MOF-5 nanocubes.** In an ice-water bath, 0.1 mmol of 1,4-benzenedicarboxylic acid was dissolved in 15 mL of N,N-dimethylformamide (DMF), which was dropwise added into 35 mL of 0.2 M Zn $(CH_3COO)_2$ DMF solution under magnetic stirring for 15 min. Subsequently, the obtained mixture was transferred into a Teflon-lined stainless steel autoclave with a capacity of 50 mL, and the autoclave was placed in a 95 °C oven for 10 h and then cooled to room temperature naturally. The resulting white precipitates were washed with DMF several times, collected by centrifugation at 4000 r.p.m. for 10 min, and dried at 80 °C in a vacuum drying oven overnight. Finally, 3D Zn-MOF-5 nanocubes were synthesized. The synthesis of ZnCo-MOF-5 was similar to that of Zn-MOF-5 except that the molar ratio of $Zn(CH_3COO)_2 \cdot 2H_2O/Co$ $(CH_3COO)_2 \cdot 4H_2O$ for preparing ZnCo-MOF-5 was 19:1, and the prepared ZnCo-MOF-5 had a pink color.

**Synthesis of 3D ZIF-7 nanoparticles.** First, 2.5 mmol of benzimidazole was dissolved in 50 mL of methanol, which was then added into 50 mL of 0.1 mM Zn $(NO_3)_2 \cdot 6H_2O$ DMF solution. The mixture was vigorously stirred at room temperature for 6 h. The white ZIF-7 nanoparticles were collected by centrifugation at 4000 r.p.m. for 10 min, washed with ethanol and water for several times, and dried in an 80 °C oven overnight.

**Synthesis of 3D ZIF-71 nanoparticles.** First, 3 mmol of 4,5-dichloroimidazole and 1 mmol of $Zn(CH_3COO)_2 \cdot 2H_2O$ were separately dissolved in 15 mL of methanol, and the two solutions were mixed at room temperature with magnetic stirring for 2 h. The precipitates were centrifuged at 4000 r.p.m. for 10 min, washed with ethanol and water several times, and dried at 80 °C overnight in an oven. White ZIF-71 nanoparticles were ultimately obtained.

**Synthesis of 2D Zn-HMT nanosheets.** First, 7 mmol of hexamethylenetetramine (HMT) and 14.25 mmol of $Zn(NO_3)_2 \cdot 6H_2O$ were dissolved in 50 mL and 30 mL of ethanol solution, respectively. The $Zn(NO_3)_2 \cdot 6H_2O$ solution was then added to the HMT solution, and white precipitates were immediately generated. The precipitates were collected by centrifugation at 4000 r.p.m. for 10 min, washed with ethanol several times, and dried in an 80 °C vacuum oven overnight, yielding Zn-HMT nanosheets.

**Synthesis of 2D ZnCo-PPF-3 nanosheets.** First, 0.015 mmol of $Zn(NO_3)_2 \cdot 6H_2O$, 0.135 mmol of $Co(NO_3)_2 \cdot 6H_2O$, 0.1 mmol of 4,4'-bipyridine and 100 mg of

polyvinylpyrrolidone (average mol wt 40,000) were dissolved in 60 mL of DMF-ethanol mixed solution (volume ratio of $V_{DMF}:V_{ethanol} = 3:1$) in a 100 mL vial. Next, 0.05 mmol of 5,10,15,20-tetrakis(4-carboxylphenyl)porphyrin was dissolved in 20 mL of DMF-ethanol mixed solution (volume ratio of $V_{DMF}:V_{ethanol} = 3:1$), which was dropwise added into the former solution and then sonicated for 25 min. Afterwards, the vial was sealed and heated to 80 °C for 24 h. The resulting red precipitates were collected by centrifugation at 8000 r.p.m. for 15 min, washed with ethanol several times, and dried at 80 °C in an oven overnight, ultimately yielding ZnCo-PPF-3 nanosheets.

**Synthesis of 1D Zn-BTC/ZnCo-BTC nanowires.** First, 2.15 mmol of Zn $(CH_3COO)_2 \cdot 2H_2O$ and 2 mmol of 1,3,5-benzenetricarboxylic acid ($H_3BTC$) were dissolved in 10 mL and 90 mL of water, respectively. The former solution was added into the latter solution at 100 °C with magnetic stirring for 5 min. The white precipitates were centrifuged at 3000 r.p.m. for 10 min, washed with ethanol several times, and dried at 80 °C in an oven overnight, yielding Zn-BTC nanowires. The synthesis of pink-colored ZnCo-BTC nanowires was similar to that of Zn-BTC nanowires except that the molar ratio of $Zn(CH_3COO)_2 \cdot 2H_2O/Co$ $(CH_3COO)_2 \cdot 4H_2O$ for preparing ZnCo-BTC was varied. XZnYCo-BTC denoted ZnCo-BTC with different molar ratios of Zn/Co, in which X and Y corresponded to the molar percentages of Zn and Co, respectively.

**Synthesis of 1D ZnCo-MOF-74 nanowires.** First, 1 mmol of Zn $(CH_3COO)_2 \cdot 2H_2O$ and 3 mmol of $Co(CH_3COO)_2 \cdot 4H_2O$ were dissolved in 10 mL of water, which was then added into 90 mL of 2,5-dihydroxy-1,4-benzenedicarboxylic acid (2 mmol) aqueous solution heated at 100 °C in an oil bath for 40 min. The earth yellow ZnCo-MOF-74 was centrifuged at 1000 r.p.m. for 10 min, washed with ethanol several times, and dried at 80 °C in an oven overnight. XZnYCo-MOF-74 denoted ZnCo-MOF-74 with different molar ratios of Zn/Co, in which X and Y corresponded to the molar percentages of Zn and Co, respectively.

**Synthesis of Zn-MOF-5-derived ZIF-8-based nanostructures.** When the prepared Zn-MOF-5 powder was placed in 1.2 M 2-methylimidazole (Hmim) ethanol solution at room temperature overnight, the transformation from Zn-MOF-5 nanocubes to single-shelled hollow ZIF-8 nanocubes (SH-ZIF-8) occurred. When the reaction time was decreased to 2 min, MOF-5@ZIF-8 yolk-shelled nanocubes could be obtained. When the reaction was conducted at 45 °C overnight, double-shelled hollow ZIF-8 nanocubes (DH-ZIF-8) were synthesized. In addition, if the concentration of Hmim ethanol solution was diluted from 1.2 M to 0.6 M during the reaction, an interesting ball-in-box ZIF-8 architecture was consequently produced. All of the products were collected by centrifugation at 3000 r.p.m. for 15 min, washed with ethanol and water several times, and dried in a 70 °C oven.

**Synthesis of ZnCo-MOF-5-derived ZnCo-ZIF nanostructures.** When the as-synthesized ZnCo-MOF-5 powder was added into 0.6 M Hmim ethanol solution for 6 h in a 0 °C icebox, single-shelled ZnCo-ZIF hollow nanocubes were synthesized. If the concentration of Hmim ethanol solution was increased to 1.2 M, double-shelled ZnCo-ZIF hollow nanocubes were prepared, and some triple-shelled ZnCo-ZIF hollow nanocubes could also be observed in the product. The resulting ZnCo-ZIF nanostructures were collected by centrifugation at 3000 r.p.m. for 15 min, washed with ethanol and water several times, and dried in a 70 °C oven.

**Synthesis of ZIF-7-/ZIF-71-derived ZIF-8 hollow particles.** When the prepared ZIF-7 was added into 2.4 M Hmim ethanol solution at 45 °C for 6 h, single-shelled ZIF-8 hollow nanoparticles could be prepared. Under the same conditions, double-shelled ZIF-8 hollow nanoparticles would be realized when the mother MOF was ZIF-71; if 1.2 M Hmim ethanol solution was used, single-shelled ZIF-8 hollow nanoparticles were obtained. The resulting ZIF-8 hollow nanoparticles were collected by centrifugation at 3000 r.p.m. for 15 min, washed with ethanol and water several times, and dried in a 70 °C oven.

**Synthesis of Zn-HMT-derived ZIF-8-based nanostructures.** When the prepared Zn-HMT was added to 0.75 M and 0.2 M Hmim methanol solution at 30 °C for 15 min, 2D ZIF-8 nanosheets and sesame pancake-like Zn-HMT@ZIF-8 nanosheets could be synthesized, respectively. The products were collected by centrifugation at 3000 r.p.m. for 15 min, washed with ethanol and water several times, and dried in a 70 °C oven.

**Synthesis of Zn-BTC-derived double-shelled ZIF-8 nanotubes.** When the as-prepared Zn-BTC nanowires were transferred into 1.2 M Hmim ethanol/water mixed solution (volume ratio of $V_{ethanol}:V_{water} = 9:1$) at 45 °C for 2 h, double-shelled ZIF-8 nanotubes (DT-ZIF-8) were synthesized. Under the same reaction conditions, when 90Zn10Co-BTC nanowires and 50Zn50Co-BTC nanowires were used as the mother MOFs, they gave rise to double-shelled ZnCo-ZIF nanotubes (DT-ZnCo-ZIF) and single-shelled ZnCo-ZIF nanotubes (ST-ZnCo-ZIF), respectively.

**Synthesis of ZnCo-BTC-derived ZIF-based nanostructures**. The preparation of ZnCo-BTC-derived ZnCo-ZIF-based nanostructures was similar to that of Zn-BTC-derived double-shelled ZIF-8 nanotubes, only 90Zn10Co-BTC nanowire was used as the mother MOF. When 90Zn10Co-BTC nanowires were transferred into 1.2 M Hmim ethanol/water mixed solution (volume ratio of $V_{ethanol}:V_{water} = 9:1$) at 45 °C for 10 min, 5 min and 1 min, 1D 90Zn10Co-BTC@ZnCo-ZIF nanowire-nanotube structures, 1D 90Zn10Co-BTC@ZnCo-ZIF core-shell nanowires and 1D 90Zn10Co-BTC@ZnCo-ZIF bead-on-string structures were produced, respectively. When placing the 90Zn10Co-BTC@ZnCo-ZIF nanowire-nanotube structure into 1.2 M Hmim ethanol/water mixed solution 45 °C for 2 h again, a peapod-like ZnCo-ZIF structure emerged. The products were all collected by centrifugation at 3000 r.p.m. for 15 min, washed with ethanol and water several times, and dried in a 70 °C oven.

**Synthesis of ZnCo-MOF-74-derived ZIF-based nanostructures**. The 20Zn80Co-MOF-74 nanowires were selected as the mother MOF. Compared to ZnCo-BTC, the 20Zn80Co-MOF-74 nanowires were placed in 1.2 M Hmim ethanol/water mixed solution (volume ratio of $V_{ethanol}:V_{water} = 5:5$) to induce the transformation. When the reaction continued for 2 h, single-shelled ZnCo-ZIF nanotubes were synthesized. When the reaction was shortened to 15 min, the resulting material evolved to a 20Zn80Co-MOF-74@ZnCo-ZIF core-shell structure. All of the products were collected by centrifugation at 1000 r. p. m. for 10 min, washed with ethanol and water several times, and dried in a 70 °C oven.

**Synthesis of solid ZIF-8 particles**. For comparison, 0.8 mmol of Zn $(CH_3COO)_2 \cdot 2H_2O$ and 0.3 g of polyvinylpyrrolidone (average mol wt 30,000) were dissolved in 20 mL of methanol and placed in a refrigerator for 1 h. In the meantime, 3.2 mmol of Hmim was dissolved in 20 mL of methanol and placed in the refrigerator for 1 h. Subsequently, the former solution was dropwise added to the latter solution at room temperature with magnetic stirring for 24 h. Solid ZIF-8 particles were collected by centrifugation, washed with ethanol and water several times, and dried in a 70 °C oven.

**Synthesis of ZIF-derived nanoporous carbon nanostructures**. The MOF precursors, including solid ZIF-8 particles, MOF-5-derived DH-ZIF-8 nanocubes, Zn-HMT-derived ZIF-8 nanosheets, Zn-BTC-derived DT-ZIF-8 nanotubes and ZnCo-BTC-derived ZnCo-ZIF nanotubes, were heated to 900 °C in an Ar atmosphere for 2 h at a heating rate of 2 °C/min. After cooling to room temperature naturally, the black powders were washed with 5 M $HNO_3$ aqueous solution at 80 °C for 24 h to remove the Zn or Co species. The resulting materials were collected by centrifugation, washed with ethanol and water several times, and finally dried in a vacuum oven at 120 °C.

**Material characterizations**. The X-ray diffraction (XRD) patterns were recorded by a Rigaku D/MAX2500VL/PC X-ray diffractometer with Cu Kα radiation. The morphology and microstructure of the samples were examined by field-emission scanning electron microscopy (SEM, SU8020) and transmission electron microscopy (TEM, JEM-2100). Nitrogen adsorption-desorption analysis was carried out by using a Micromeritics ASAP 2020 instrument at 77 K, and the surface area values were calculated by the Brunauer-Emmett-Teller (BET) method in the relative pressure $(P/P_o)$ range of 0.002–1.0. X-ray photoelectron spectroscopy (XPS) was performed by a Thermo Fisher X-ray photoelectron spectrometer (ESCALAB250), and the results were fitted using XPSpeak software according to the principle of minimum residual standard deviation. The Raman spectrum was collected from an Ar laser (Renishaw inVia) with a 532 nm laser excitation source at room temperature. The Fourier transform infrared spectrum (FT-IR) was collected on a Perkin Elmer Fourier-Transform infrared spectrometer by using the KBr wafer technique.

**Electrochemical measurements**. The electrochemical properties of ZIF-derived nanoporous carbon nanostructures were estimated by assembling 2032 type coin cells. To prepare the working electrodes, the active materials were mixed with Super P® carbon black and polyvinylidene fluoride at a weight ratio of 80:10:10 in N-methyl pyrrolidinone (NMP) to make a slurry mixture, which were pasted on copper foil and dried in a vacuum oven at 60 °C for 12 h and then 120 °C for another 2 h. The mass loading of the working electrodes was 0.8 ~ 1.0 mg/cm². Metal Na foil was used as the counter electrode, and borosilicate glass microfiber (GF/D, Whatman) was used as the separator. The electrolyte was 1 M $NaClO_4$ in a mixed solution of ethylene carbonate (EC) and dimethyl carbonate (DMC) (volume ratio of $V_{EC}:V_{DMC} = 1:1$). The coin cells were assembled in an Ar-filled glovebox (MIKROUNA) and allowed to rest overnight before the electrochemical tests. The galvanostatic discharge/charge profiles were recorded on a Land CT2001A battery tester (China) in a 25 °C thermostat (FYL-YS-280L, Beijing Fu Italian Electric, China). The cyclic voltammetry analysis and electrochemical impedance spectroscopy (EIS) measurements were conducted on a CHI660C electrochemical workstation (Chenhua Instrument Company, China). EIS spectra were recorded in a frequency range of 0.1 Hz ~100 kHz at the open-circuit potential with an ac perturbation of 0.1 V.

## Data availability

The authors declare all data supporting the findings of this study are available within the paper and the Supplementary Information file, or available from the corresponding authors on reasonable request. The source data underlying all figures are provided as a Source Data file.

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

## Acknowledgements

We acknowledge the funding support from the National Natural Science Foundation of China (Nos. 21606217, U1810204 and 91534203), Fundamental Research Funds for the Central Universities (No. PA2019GDQT0015) and the 111 Project "New Materials and Technology for Clean Energy" (B18018). J.W.C. would also like to thank the financial support from the China Scholarship Council during his visit to Prof. Pulickel M. Ajayan's group at Rice University.

## Author Contributions

D.B.Y., J.W.C., T.W.X. and Y.C.W. conceived the project. D.B.Y., Q.S., Q.J.S. and Y.L.Z. planned and performed the experiments and collected and analyzed the data. B.W., L.G., Y.W., Y.Z., Y.Q.Q., H.T.W. and P.M.A. assisted with the experiments and characterizations. D.B.Y., J.W.C., T.W.X., R.V. and Y.C.W. co-wrote the manuscript. All authors discussed the results and commented on the manuscript. D.B.Y. and Q.S. contributed equally to this work.

## Competing interests

The authors declare no competing interests.
