## [Peer Review File · Nature Communications]

Reviewers' comments:

Reviewer #1 (Remarks to the Author):

The manuscript "A Solvent-Assisted Ligand Exchange Approach Enables Metal-Organic Frameworks with Diverse and Complex Architectures" by Wu and coworkers is an excellent report on the formation of complex MOF structures. It is well-written and should be published. Nevertheless, I have some small comments that should be addressed.

1. Is it possible to study the kinetics of ligand exchange/MOF transformation in more detail? Maybe by watching the change in time in SEM and XRD after quenching?
2. What is the effect of initial imidazole concentration on the process?
3. There are some reports in literature the authors should take a look at (especially the first one):
<https://doi.org/10.1002/slct.201600526>
<https://pubs.acs.org/doi/abs/10.1021/jacs.7b12633>
<https://onlinelibrary.wiley.com/doi/full/10.1002/anie.201901707>
<https://pubs.acs.org/doi/full/10.1021/ja403810k>
<https://pubs.acs.org/doi/10.1021/acs.cgd.8b01628>
4. The authors showed some examples about concentration dependency. Nevertheless, I miss some summarizing discussion/rationale behind the study.
5. Why are double shelled structures formed instead of direct growth for a thicker shell?
6. The evaporation of Zn species during carbonization can also have an effect on porosity and should be discussed.
7. Small errors in writing: "nanowires was"; "peapod-like MOF structures have been reported before" (not been reported?)
8. The authors should add page numbers to the SI and a table of contents

Reviewer #2 (Remarks to the Author):

In this paper, the author reported new, facile and versatile synthesis strategies for the design of diverse and complex MOF architectures. By precisely adjusting the balance between the cleavage rate of old bonds and the formation rate of new bonds, 21 different MOF materials and 13 different types of MOF configurations ranging from 3D to 2D and 1D structures have been successfully prepared. In addition, after a carbonization treatment, MOF-derived nanoporous carbon was used for Na⁺ ion storage. Considering the overall quality of the paper, I recommend its publication after minor revision.

1. The author mentioned that "In order to demonstrate the advantages of MOF architectures, the structured MOFs are subsequently carbonized to fabricate nanoporous carbon for Na⁺ ion storage" in the last paragraph of the introduction, however, it seems illogical. In view of the advantages of MOF architectures, why not the author uses the structured MOFs directly for Na⁺ ion storage? After all, no introduction about the advantage of carbonization was mentioned in the preamble.
2. In line 132, is there something wrong with the "Hmim H⁺ + mim⁻". Please check it carefully.
3. In line 322, "CV curves of all ZIF-derived porous carbon are measured at various scan rates", it's better to put the related figure into the main manuscript.
4. Please check the captions of Fig. 2 and 3, I think it's inappropriate to begin with "Preparation of ...".
5. Although there are a large number of references with good quality, some important articles should be cited. e.g: Coordination Chemistry Reviews, 2019, 389: 119-140; Advanced Materials, 2019,

31(6): 1804740; Coordination Chemistry Reviews, 2018, 376: 292-318; Advanced Functional Materials, 2018, 28(47): 1804950.

Reviewer #3 (Remarks to the Author):

As one of the main limitations for a broader application of MOF, controllable preparation of MOF with diverse and complex micro/nano-structures is still at a very early stage. In this paper, the authors developed a powerful synthesis strategy called "SALE" approach, and synthesized 13 different MOF structures including a kind of rare double-shelled nanotube. These structured MOF materials were carbonized to nanoporous carbon further, which exhibited excellent electrochemical property for Na⁺ storage. Overall, this work is very interesting and would add significant contribution to this field, the manuscript is well-organized, and the authors have done relatively detailed investigations and discussions. Therefore, I strongly recommend the publication of this work in Nature Communications after a few clarifications.

1. It would be better to include FTIR spectra characterization for ZIFs after the SALE process.
2. The stability of coordination bonds for ZIF-7/ZIF-71 corresponded to pKa value of linkers, how about Zn-HMT/ZnCo-PPF-3 and ZnCo-BTC/ZnCo-MOF-74 ?
3. In recent publications MOF-derived single-atom catalysts often show 3D solid particle shape, do you think whether or not single-atom catalysts derived from the prepared MOF nanostructures by your approach have an advantage over them ?

Response to Referees

Dear Referees,

Thank you very much for all your comments on our manuscript entitled “A Solvent-Assisted Ligand Exchange Approach Enables Metal-Organic Frameworks with Diverse and Complex Architectures” (Manuscript ID: NCOMMS-19-28948-T). Your insightful comments and suggestions on our manuscript are all valuable and very helpful for us to improve the quality of our paper. We have now revised the whole manuscript carefully. The changes are marked in red font in the revised manuscript and supplementary information. Our response to your comments point by point is listed below.

Referee 1

The manuscript "A Solvent-Assisted Ligand Exchange Approach Enables Metal-Organic Frameworks with Diverse and Complex Architectures" by Wu and coworkers is an excellent report on the formation of complex MOF structures. It is well-written and should be published. Nevertheless, I have some small comments that should be addressed.

Reply: We thank you very much for your positive remarks on our work, we really appreciate it.

1. Is it possible to study the kinetics of ligand exchange/MOF transformation in more detail? Maybe by watching the change in time in SEM and XRD after quenching?

Reply: The suggestion is highly helpful. We first tried to capture the states of transformation process in 1.2 M 2-methylimidazole (Hmim) ethanol solution, and found that the kinetics was too fast, although we realized that a core-shell structure would be an intermediate state before forming yolk-shell structure. In order to get more detailed information of the transformation as a function of time, 0.3 M Hmim was used to slow down the transformation kinetics, the nucleation of ZIF-8 nanocrystals was clearly seen on the surface of MOF-5, and ZIF-8 was gradually covering on the facets of MOF-5 nanocubes, resulting in the formation of yolk-shell

ZIF-8@MOF-5 hybrid structure and finally the generation of ball-in-box structure. About the related discussions on the evolution, please kindly see Supplementary Figure 5 and Page 7 in the revised manuscript.

2. What is the effect of initial imidazole concentration on the process?

Reply: We have done the related experiments to investigate the influence of initial Hmim concentration on the nanostructures of final products (Supplementary Figure 5, Supplementary Figure 9). We find out that a proper concentration range is needed not only to drive a rational kinetics for crystallizing ZIF-8, but also to offer steady ligand exchange for inheriting the shape of MOF-5 nanocubes. In addition, a higher concentration facilitates faster kinetics for phase separation in a shorter time, yielding a yolk-shell structure. The ligand concentration also determines the mim^- concentration gradient along the vertical direction of shell wall, which would further affect $\text{mim}^-/\text{Zn}^{2+}$ ratio inside the hollow interior and thus the evolution behavior of MOF-5 core. It is noted that in the case of ZIF-71-derived double-shelled hollow ZIF-8, the higher mim^- concentration would enable the sufficient mim^- ligands inside the hollow interior for crystallizing ZIF-8, leading to the formation of secondary ZIF-8 shell (Pages 8-9 in the revised manuscript).

3. There some reports in literature the authors should take a look at (especially the first one):

[\(a\) https://doi.org/10.1002/slct.201600526](https://doi.org/10.1002/slct.201600526)

[\(b\) https://pubs.acs.org/doi/abs/10.1021/jacs.7b12633](https://pubs.acs.org/doi/abs/10.1021/jacs.7b12633)

[\(c\) https://onlinelibrary.wiley.com/doi/full/10.1002/anie.201901707](https://onlinelibrary.wiley.com/doi/full/10.1002/anie.201901707)

[\(d\) https://pubs.acs.org/doi/full/10.1021/ja403810k](https://pubs.acs.org/doi/full/10.1021/ja403810k)

[\(e\) https://pubs.acs.org/doi/10.1021/acs.cgd.8b01628](https://pubs.acs.org/doi/10.1021/acs.cgd.8b01628)

Reply: We thank Referee 1 for providing a list of relevant literature, and we have carefully studied these useful papers especially Ref. (a). Ref. (a) demonstrates a MOF-to-MOF

conversion strategy to synthesize ZIF-8 hollow spheres derived from MOF-5 cubes. The preparation of ZIF-8 hollow spheres was conducted at 120 °C. When MOF-5 was added into a 20 mM of Hmim DMF solution, according to our study, the very high temperature might be responsible for the loss of cubic shape because it enables the quick dissolution of MOF-5 (cleavage of Zn-O bond), and no substrate is available for ZIF-8 nucleation (establishment of Zn-N bond). The effects of Hmim concentration and reaction time were investigated and discussed in this paper, they found out that amorphous Zn-imidazolate coordination compounds were formed at the beginning which served as the ingredient of ZIF-8 shell, and the shell wall became thicker with the increase of reaction time; the low concentration resulted in a yolk-shell structure (the yolk was composed of the fused particulates) and an incomplete conversion of Zn-O to Zn-N. These phenomena accorded with the observations of our MOF-5-derived ZIF-8 nanostructures (Supplementary Figure 5, Supplementary Figure 9). Ref. (b) reported a double hydrophilic block copolymer-modulated method to design a core-shell MOF hybrid with relatively low crystallinity of MOF shell. In Ref. (c), multilayer MOF-on-MOF films were fabricated by epitaxial growth owing to negligible lattice mismatch among these MOFs. Ref. (d) and Ref. (e) are papers about the functional modification of MOFs by ligand exchange strategy. We have now cited these papers as Ref. 28, Ref. 29, Ref. 34, Ref. 35, Ref. 45 in the revised manuscript.

4. The authors showed some examples about concentration dependency. Nevertheless, I miss some summarizing discussion/rationale behind the study.

Reply: Thanks for the Comment 2 from the Referee 1, we have summarized the effect of ligand concentration on the nanostructures of final products. Please see the Reply to Comment 2 or Page 9 in the revised manuscript.

5. Why are double shelled structures formed instead of direct growth for a thicker shell?

Reply: In our case, the occurrence of phase separation to be a yolk-shell-like structure is the necessary condition for preparing double-shelled structures, which is originated from the non-equivalent diffusion during the ligand exchange process. On the other hand, a certain molar ratio of diffused $\text{mim}^-/\text{Zn}^{2+}(\text{Co}^{2+})$ inside the hollow interior is needed to proceed the crystallization of secondary ZIF shell evolved from yolk mother MOFs. Otherwise, due to the trend of reducing nucleation energy and the relatively higher $\text{mim}^-/\text{Zn}^{2+}$ ratio nearby the shell wall than in the center, the nucleation of ZIF prefers to continue on the inner wall of outer shell with the continuous dissolution of mother MOFs, resulting in the formation of single-shelled hollow structures (Page 8 in the revised manuscript). Of course, theoretically, there is another possibility of forming single-shelled structures that the phase separation never happens during the whole process, when the vacancies have enough time to diffuse into the center rather than accumulate at the phase interface (in our latest experiments, we observe such structure evolution from ZIF-71 to its derived single-shelled hollow ZIF-8, we never demonstrate the detailed results in this manuscript but in our following study). Considering the big significance of preparing 1D MOF nanostructures, to demonstrate it more clearly, we also add the discussions about the formation of single-/double-shelled nanotube structures in detail as shown in Pages 12-13 in the revised manuscript.

6. The evaporation of Zn species during carbonization can also have an effect on porosity and should be discussed.

Reply: In the revised manuscript (Page 16), we have discussed the formation of micro-/meso-/macro-pores of ZIF-derived carbon during the carbonization process.

7. Small errors in writing: "nanowires was"; "peapod-like MOF structures have been reported before" (not been reported?)

Reply: We have carefully checked and corrected the possible errors within the revised manuscript.

8. The authors should add page numbers to the SI and a table of contents.

Reply: Thanks. We have added the page numbers and a table of contents in Supplementary Information.

Referee 2

In this paper, the author reported new, facile and versatile synthesis strategies for the design of diverse and complex MOF architectures. By precisely adjusting the balance between the cleavage rate of old bonds and the formation rate of new bonds, 21 different MOF materials and 13 different types of MOF configurations ranging from 3D to 2D and 1D structures have been successfully prepared. In addition, after a carbonization treatment, MOF-derived nanoporous carbon was used for Na⁺ ion storage. Considering the overall quality of the paper, I recommend its publication after minor revision.

Reply: We thank you very much for your positive remarks on our work, we really appreciate it.

1. The author mentioned that “In order to demonstrate the advantages of MOF architectures, the structured MOFs are subsequently carbonized to fabricate nanoporous carbon for Na⁺ ion storage” in the last paragraph of the introduction, however, it seems illogical. In view of the advantages of MOF architectures, why not the author uses the structured MOFs directly for Na⁺ ion storage? After all, no introduction about the advantage of carbonization was mentioned in the preamble.

Reply: Thanks for the valuable comment. In fact, it has been reported that some MOF materials can be directly used for electrochemical applications, but there is a serious concern that MOFs like ZIFs often suffer from high ohmic resistance and poor stability when assembled into practical devices, leading to very low electrochemical performance. In contrast, MOF-derived nanoporous carbon by an optimized carbonization treatment can combine the excellent conductivity and stability of carbon with the porous feature and nano-architectures inherited from pristine MOF precursors, which is expected to exhibit high-efficiency performance. Hence, we have now added the related discussion on why to use MOF-derived carbon for Na⁺ ion storage rather than pristine MOFs into the Introduction Part (Page 5 in the revised manuscript).

2. In line 132, is there something wrong with the “Hmim \$\rightleftharpoons\$ H⁺+mim⁻”. Please check it carefully.

Reply: Yes. 2-methylimidazole (Hmim) actually co-exists in the deprotonated state as a linker unit and in the neutral state as a stabilizing unit, therefore, a more scientific formula should be $\text{Hmim} \rightleftharpoons (1-x)\text{H}^+ + (1-x)\text{mim}^- + x\text{Hmim}$, now we have corrected it (Page 7 in the revised manuscript).

3. In line 322, “CV curves of all ZIF-derived porous carbon are measured at various scan rates”, it’s better to put the related figure into the main manuscript.

Reply: We have added the related CV curves in Figure 4.

4. Please check the captions of Fig. 2 and 3, I think it’s inappropriate to begin with “Preparation of ...”.

Reply: Thanks. The captions of Figure 2 and Figure 3 have been changed, please see Pages 28-29 in the revised manuscript.

5. Although there are a large number of references with good quality, some important articles should be cited. e.g: Coordination Chemistry Reviews, 2019, 389: 119-140; Advanced Materials, 2019, 31(6): 1804740; Coordination Chemistry Reviews, 2018, 376: 292-318; Advanced Functional Materials, 2018, 28(47): 1804950.

Reply: We have cited all the papers in the revised version (Ref. 6, Ref. 11, Ref. 38, Ref. 39).

Referee 3

As one of the main limitations for a broader application of MOF, controllable preparation of MOF with diverse and complex micro/nano-structures is still at a very early stage. In this paper, the authors developed a powerful synthesis strategy called “SALE” approach, and synthesized 13 different MOF structures including a kind of rare double-shelled nanotube. These structured MOF materials were carbonized to nanoporous carbon further, which exhibited excellent electrochemical property for Na⁺ storage. Overall, this work is very interesting and would add significant contribution to this field, the manuscript is well-organized, and the authors have done relatively detailed investigations and discussions. Therefore, I strongly recommend the publication of this work in Nature Communications after a few clarifications.

Reply: We thank you very much for your positive remarks on our work, we really appreciate it.

1. It would be better to include FTIR spectra characterization for ZIFs after the SALE process.

Reply: We have recorded the FT-IR spectra of all prepared ZIF-based nanostructures (Supplementary Figure 2). The results clearly demonstrated the successful coordination between 2-methylimidazole and Zn, suggesting the transformation of ZIFs from those mother MOFs after the SALE process (Page 6 in the revised manuscript).

2. The stability of coordination bonds for ZIF-7/ZIF-71 corresponded to pKa value of linkers, how about Zn-HMT/ZnCo-PPF-3 and ZnCo-BTC/ZnCo-MOF-74?

Reply: The metal-H₂O and hydrogen bonding interaction are involved to construct the frameworks of both Zn-HMT and ZnCo-BTC, while the metal ions in ZnCo-PPF-3 and ZnCo-MOF-74 all coordinate to organic ligands whose bonding is much stronger. Therefore, our experimental results demonstrate that it is difficult to use ZnCo-PPF-3 nanosheet and ZnCo-MOF-74 for preparing ZnCo-ZIF nanosheets and ZnCo-ZIF nanotube, respectively. The detailed discussions are presented in Supplementary Figure 20 and Supplementary Figure 31.

3. In recent publications MOF-derived single-atom catalysts often show 3D solid particle shape, do you think whether or not single-atom catalysts derived from the prepared MOF nanostructures by your approach have an advantage over them?

Reply: We believe structured MOF nanomaterials will also show promise for the application in single-atom catalysts. For example, the overall CO₂ reduction reaction actually involves three steps, while each atomically dispersed transition metal (e.g. Ni, Fe, Co) anchored on N-doped carbon often has sole catalytic activity for one of the steps. Designing a kind of core-shell hybrid MOF structure would be an alternative solution of this concern, the core MOF-derived and the shell MOF-derived carbon can respectively offer different functionalities in order to realize multistep catalytic reactions. Moreover, 1D/2D nanostructured catalysts are known to have faster electron transfer and more accessible

molecular/ion diffusion path than 3D solid particle-shape counterparts. Therefore, our prepared 1D/2D MOF nanostructures would have great potential application in single-atom catalysts as well.

REVIEWERS' COMMENTS:

Reviewer #1 (Remarks to the Author):

The revised version of the manuscript "A Solvent-Assisted Ligand Exchange Approach Enables Metal-Organic Frameworks with Diverse and Complex Architectures" by Ajayan and coworkers addresses all my previous comments. It should be published in Nature Communications as is.

Reviewer #3 (Remarks to the Author):

The manuscript have been revised carefully according to the comment, and I recommend it for publication.

Response to Referees

Dear Referees,

Thank you very much for your positive comments on our manuscript entitled “**A solvent-assisted ligand exchange approach enables metal-organic frameworks with diverse and complex architectures**” (*Nature Communications-Manuscript ID NCOMMS-19-28948A*). We really appreciate your suggestions on our manuscript. Our responses to your comments are listed below.

Reviewer #1 (Remarks to the Author):

The revised version of the manuscript "A Solvent-Assisted Ligand Exchange Approach Enables Metal-Organic Frameworks with Diverse and Complex Architectures" by Ajayan and coworkers addresses all my previous comments. It should be published in Nature Communications as is.

Reply: We thank very much for your positive remarks on our work, we really appreciate it.

Reviewer #3 (Remarks to the Author):

The manuscript have been revised carefully according to the comment,and I recommend it for publication.

Reply: We thank very much for your positive remarks on our work, we really appreciate it.